# The Effect of the Wild Boar Hunting System on Agricultural Damages: The North-East of Italy as a Case Scenario

**DOI:** 10.3390/ani14010042

**Published:** 2023-12-21

**Authors:** Valentina Cecchini, Marcello Franchini, Michele Benfatto, Stefano Filacorda, Mirco Corazzin, Stefano Bovolenta

**Affiliations:** 1Department of Agrifood, Environmental and Animal Sciences, University of Udine, Via delle Scienze 206, 33100 Udine, Italy; valentina.cecchini@uniud.it (V.C.); stefano.filacorda@uniud.it (S.F.); mirco.corazzin@uniud.it (M.C.); stefano.bovolenta@uniud.it (S.B.); 2Hunting Service and Fishing Resources, Friuli Venezia Giulia Region, Via Sabbadini 31, 33100 Udine, Italy; michele.benfatto@regione.fvg.it

**Keywords:** agro-livestock system, crop, human–wildlife conflict, official claims, *Sus scrofa*

## Abstract

**Simple Summary:**

The wild boar *Sus scrofa* is considered as one of the main agricultural pests by farmers. Hunting is a common strategy used across Europe to reduce wild boar abundance and, in turn, damages to agricultural crops. However, results are still debated. In this research, data on official claims (i.e., damages to crops), wild boar local counts and hunting bags collected from 2019 to 2022 were analysed to ascertain the effect of the hunting system carried out in the north-east of Italy on the number of crop damages. The findings obtained showed no clear spatial overlap among wild boar hunting, wild boar density and damages to agriculture. Furthermore, the current hunting system did not produce significant effects on crop damages by wild boar. The numeric control of the species oriented towards more age classes in a similar percentage should be preferred to mitigate damages.

**Abstract:**

Hunting is a method commonly used in several European countries to reduce crop damages by wild boar *Sus scrofa*. However, results are still controversial and poorly treated. Using data on official claims (i.e., damages to crops) and wild boar local counts and hunting bags collected from 2019 to 2022, the purpose of this work was to evaluate the effect of the hunting system (divided into eradication and non-eradication areas) carried out in the north-east of Italy on the number of agricultural damages. The spatio-temporal distribution of wild boar hunting, density and damages as well as the effect of hunting, were evaluated through the hotspot analysis and the zero-inflated models, respectively. The results obtained revealed no clear spatial overlap among wild boar hunting, wild boar density and damages to agriculture in both the eradication and non-eradication areas. Moreover, the current level of harvesting did not significantly affect the number of agricultural damages. A multifaceted approach focused on the numeric control of the species based on accurate local counts and oriented towards more age classes in a similar percentage should be preferred to mitigate damages to cultivars.

## 1. Introduction

The wild boar *Sus scrofa* mostly prefers mixed environments including forests and wetlands [1,2,3]. However, its remarkable ecological plasticity has allowed the species to expand its distribution range, both in native areas and introduced ranges [4,5]. In Europe, the species is present in several countries [6], except for some northern areas characterized by harsh climate conditions [7]. Several authors concur that the European wild boar population started to increase in the second half of 1960s and has still been showing an increasing trend [8,9,10]. The wild boar population growth was influenced by several factors, including reforestation, abandonment of agricultural areas, national legislations, development of wildlife-management institutions, and active (re)introduction for hunting purposes [11,12,13]. Furthermore, the expansion of the species was favoured by the recovery of forests [2,3], its high reproductive potential [14,15], effects of climate change [14,16,17], absence or low densities of natural predators [18], supplemental feeding [7], and adaptation to human-altered environments [13,19]. In Italy, the wild boar is widespread across the entire peninsula, both in lowlands and mountainous areas [20]. The current available literature reported an increasing population trend across time, spanning from 600,000 individuals estimated in the period 1996–2000 [21] and up to 1.5 million in 2021 [20].

The wild boar is considered one of the main wildlife species causing agriculture damages. In fact, until the middle of the last century, the wild boar was mainly concentrated in forests, while, in recent decades, the species also started to exploit human-altered environments (e.g., peri-urban areas and agricultural lands) and grasslands [9,22,23,24,25]. Damages caused by wild boar are essentially linked to food search, as well as physiological and ethological needs [24,26,27]. Crop damages can occur in periods in which food resources are scant in natural habitats [24]. Specifically, the odds of damages increase in those areas in which crops are in the near proximity of forest and shrub habitats [27,28,29], and/or where hunting is forbidden [30].

Although their efficacy is questionable and varies depending on the local context, some prevention measures used to reduce crop damages are fencing and supplementary feeding [31,32], sterilization [32,33] and hunting [31,32]. Hunting can be used as a tool for wild boar management [34] to prevent agricultural damages [28] and limit population growth [7,35], and its effect is linked to several factors such as the number of hunters (in decline) [9], wild boar population size, number of hunting days (influenced by local legislation [34,36]) and number of hunting dogs used during the hunting operations [9,30]. When wild boar became highly abundant (especially within introduced ranges), beyond causing damages to crops, they can produce negative effects on biodiversity, cause road accidents and transmit infectious diseases to both humans and livestock [34]. Therefore, a sustainable and well-planned hunting system focused on reducing the magnitude of these negative effects, and, at the same time, preserving the wild boar populations in their native ranges may represent a solution [31,34]. The effect of hunting in mitigating wild boar damages to agricultural fields is still poorly treated across European countries (e.g., [24,29,31,34,37]), and the findings obtained are controversial. For instance, a study carried out in Switzerland [31] revealed that among the three methods tested to reduce the agricultural damages by wild boar (i.e., hunting, supplementary feeding in forests, electric fences around crops) only hunting significantly reduced damages. Conversely, research in north-eastern Italy [24] showed a non-significant relationship between the number of shot wild boar and the amount of damages to cultivars. Moreover, it was also shown that the current level of harvesting was ineffective in reducing the wild boar population growth, likely because of the species’ high reproductive potential triggered by a density-dependent compensation effect [3,24]. Given these inconsistencies, more studies are needed to provide further insights.

Based on these considerations, the purpose of this work was to evaluate the effect of the hunting system carried out in the north-east of Italy on the number of agricultural damages. To achieve this goal, we asked the main following research questions: (*i*) Does the current wild boar hunting rate affect the overall number of agricultural damages? (*ii*) Does the current hunting rate affect the wild boar population growth across time? Given the high reproductive rate of the species affected by the density-dependent compensation effect over time, we assumed that the current hunting rate would significantly affect the population dynamic of wild boar.

## 2. Materials and Methods

### 2.1. Study Area and Data Collection

The research was carried out in the Friuli Venezia Giulia (hereafter, FVG) region, in north-eastern Italy (Figure 1). As reported in the wildlife-management hunting plan [38], within the region the species occupies Alpine and pre-Alpine territories (at higher densities [38,39]), where the habitat is mainly composed of broad-leaved (alder *Alnus* spp., beech *Fagus* spp., birch *Betula* spp., elm *Ulmus* spp., lime *Tilia* spp., maple *Acer* spp., oak *Quercus* spp.) and coniferous (larch *Larix* spp., pine *Pinus* spp., spruce *Abies* spp.) forests, and lowland areas, mainly covered by agriculture. The overall wild boar density across the Region is of about 1.7 ind./100 ha [40].

Data referring to official claims due to wild boar damages to agriculture, local counts and hunting bags of the species collected from 2019 to 2022 by hunters and wildlife technicians working for the FVG region were analysed. Data included the following information: (*i*) municipality, (*ii*) hunting reserve (in which hunting is conducted), (*iii*) area (i.e., eradication, non-eradication), (*iv*) agro-sylvo-pastoral surface (ha—surface in which wild boar are counted during counts conducted by local hunters within each hunting reserve falling within each municipality), (*v*) number of wild boar counted, (*vi*) number of wild boar hunted (during the hunting season and by way of derogation regime), (*vii*) number of official claims to obtain compensations for crop damages, and (*viii*) affected crop category.

The FVG Region provides funds for the compensation and prevention of damages caused by wildlife to vehicles, and both agricultural and livestock practices according to the Articles 10 and 39 of the Regional Law n. 6 of 6 March 2008. After an *in-situ* inspection conducted by trained personnel, both the type of damage as well as the responsible species were determined. The information referring to damages to crops, claims for compensation, and number of wild boar counted and hunted during hunting operations was obtained from the official website of the FVG region [40] and INFOFAUNA FVG database [41].

### 2.2. Wild Boar Hunting System in FVG

In FVG, the wild boar hunting system includes two types of hunting methods: traditional (i.e., collective hunting (mainly driven hunting using hunting hounds) carried out from 1 September to 31 December, for a maximum of 90 days) and selective (i.e., individual hunting conducted from raised hides carried out from 15 May to 15 January), and foresees a higher hunting pressure on juveniles (50–60%) and lower on adults (10–25%). Both hunting methods, traditional and selective, are conducted within the eradication and non-eradication areas (Figure 1) [38]. The eradication area includes municipalities in which cultivated fields are predominant (mainly in lowlands and hilly areas). In this area, the target density for the species is zero, which means that the culling plans are not strictly linked to local counts and do not have limits in terms of numbers or structure. However, the harvesting of adult females can only occur after the removal of all accompanying piglets [38]. Within the non-eradication area, which includes mainly mountainous territories, hunting plans are based on the results obtained from local counts. These are conducted from February to May at sunset and during the night on raised hides or in the near proximity of natural feeding points [38].

In both the eradication and non-eradication areas, the European Legislations (Art. n. 9 of the ‘Birds’ Directive 2009/147/EEC; Art. n. 16 of the ‘Habitat’ Directive 92/43/EEC) allows hunting by way of derogation regime in those situations in which the implementation of prevention measures was not effective and/or in those cases in which damages are consistent. Hunting by way of derogation regime is mainly carried out by members of the Forestry Service and can take place in locations and times in which regular hunting is forbidden.

### 2.3. Data Analysis

#### 2.3.1. Spatial Analysis

To explore the intensity of wild boar hunting, abundance and official claims at a municipal level, hotspot analysis [42,43,44,45] was carried out using the Hot Spot Analysis (Getis–Ord Gi*) function in ArcMap (v. 10.8, ESRI—2020) [46]. For wild boar hunted in 2019, 2020 and 2021, all the protected areas in which hunting is forbidden were excluded from the analysis, i.e., urban areas, road network, closed-end funds, protected oases, Regional Natural Parks and Reserves, State Natural Reserves, mountain passes, refuge areas, and reproduction, rewilding, and capturing areas [38]. The surface covered by urban areas was extracted from ‘Carta Natura’ FVG 2021 [47], while those referring to the remaining non-hunting areas were downloaded from the Eagle FVG Official Platform [48]. Thereafter, using Google Satellite in QGIS (v. 3.28) [49], the newly created shapefile representing the hunting areas was manually screened to remove the remaining areas (not reported in the original shapefile) in which hunting is forbidden (e.g., beaches, shooting ranges).

The hotspot analysis requires that the size of the polygons needs to be taken into account in order to obtain reliable estimates [42,43]. Therefore, the analysis was carried out considering wild boar density (i.e., the overall number of wild boar counted divided by the agro-sylvo-pastoral surface of the hunting reserve/s falling within each municipality), the hunting density (i.e., overall number of wild boar hunted divided by the surface covered by the hunting area falling within each municipality), and the official claims’ density (i.e., overall number of official claims per municipality divided by the surface covered by each habitat potentially damaged by wild boar, i.e., arboriculture (broad-leaved and coniferous plantations), intensive and extensive cultivations, orchards, vineyards and olive groves, grasslands). Because the shapefile of the agro-sylvo-pastoral surface in which wild boar counts are conducted (which represents a smaller portion of the hunting area) was not available, the hotspot map of wild boar density covered the whole hunting area (see Results). Moreover, especially for the non-eradication area, because the surface covered by the habitats potentially damaged by wild boar was reduced, to improve clarity, the hotspot map of official claims between areas (eradication, non-eradication) covered each municipality (see Results). In the hotspot analysis we compared hotspot clusters of wild boar hunting density in 2019, 2020 and 2021 with both hotspot clusters of wild boar density and official claims’ density in 2020, 2021 and 2022, respectively. Hotspot clusters of both wild boar and official claims’ densities at time t (e.g., 2020) were compared with hotspot clusters of wild boar hunting density at time t − 1 (e.g., 2019) to explore if hunting exerted a significant effect on the spatio-temporal distribution of wild boar (in turn affecting the density of official claims), taking into account the high reproductive potential of the species across time triggered by a density-dependent compensation effect [3,24].

Before implementing the hotspot analysis, both (i) the spatial autocorrelation and (ii) the incremental spatial autocorrelation of the dataset were assessed. The spatial autocorrelation was checked through the Spatial Autocorrelation (Morans I) function (which relies on the Moran’s Index), implemented in the ‘spatial statistics’ toolbox [43]. The incremental spatial autocorrelation was measured through the Incremental Spatial Autocorrelation function and, as a starting distance, the average distance (m) among layers calculated through the Calculate Distance Band from Neighbor Count function was used. Both functions were implemented in the ‘spatial statistics’ toolbox [43]. If the incremental spatial autocorrelation revealed the presence of peaks, the distance reported by the maximum peak was used as a threshold distance in the hotspot analysis [50,51].

The FVG ‘Carta Natura’ shapefile of 2021 [47] was used to extract the surface covered by each habitat within the municipalities falling within the eradication and non-eradication area. Habitats were reclassified using the case function (implemented within the Expression Dialog of the attribute table of the ‘Carta Natura’ shapefile of 2021) and then aggregated using the dissolve geoprocessing tool (implemented in QGIS). A new habitat classification was then obtained: ‘Agriculture’, ‘Broad-leaved forests’, ‘Coniferous forests’, ‘Dunes, rocky/glaciers and coastal areas’, ‘Grasslands’, ‘Mixed forests’, ‘Moors, shrublands and riparian vegetation’, ‘Urban areas’ and ‘Wetlands and water bodies’. The percentage of landscape (PLAND) was then calculated for each new habitat classification (Table 1).

#### 2.3.2. Statistical Analysis

Statistical analysis was carried out using R Software (v. 4.3.1, [52]) and setting the level of significance at 0.05.

Both chi-squared and Fisher’s exact tests [53] were used to (i) explore variation in terms of the overall number of wild boar hunted from 2019 to 2021; (ii) assess variation in terms of the overall number of wild boar counted in each hunting reserve falling within each municipality from 2020 to 2022; (iii) explore variation in terms of the overall number of official claims from 2020 to 2022; and (iv) compare percentages of damages in the different crop categories. The decision to use percentages was taken because, in the original data set, if more crops were damaged during a single event, no discernment in terms of number of official claims was completed for each affected crop category. The pairwise nominal independent function (pnif), implemented in the ‘rcompainon’ R package [54], was thereafter used for pairwise comparisons between years and/or crop categories.

We used three Generalized Linear Models (GLMs) [55] per area (eradication and non-eradication) with residuals showing a negative binomial distribution to test the effect of hunting at time t (i.e., from 2019 to 2021—covariate) on wild boar abundance at time t + 1 (i.e., from 2020 to 2022—response variable), respectively. As in the hotspot analysis, the objective was to test if the high reproductive potential of the species would have been triggered by a density-dependent compensation effect over time [3,24]. The negative binomial distribution was used to reduce the potential effect of overdispersion among data. The latter was checked by dividing the residual deviance of the original model (with residuals showing a Poisson distribution) with the respective degrees of freedom [55].

To account for the high number of zeros (i.e., municipalities in which no official claims were reported), the relationship between the number of official claims from 2020 to 2022 (response variable) and some covariates, i.e., surface covered by (i) arboriculture, (ii) extensive cultivations, (iii) intensive cultivations, (iv) orchards, vineyards and olive groves, (v) grasslands, as well as (vi) number of wild boar hunted from 2019 to 2021, and (vii) wild boar abundance from 2020 to 2022 was explored using three sets of zero-inflated models (ZIMs [56]) per area, with residuals showing a negative binomial distribution. To avoid model overfitting due to the relatively low number of samples and high number of covariates, each set of zero-inflated models included two different models: one testing the effect of the habitat covariates on official claims, and the other testing the effect of wild boar hunting and abundance on the same response variable. The function used to launch the models was zeroinfl, implemented in the ‘pscl’ R package [57]. The negative binomial distribution of residuals was chosen to best fit with over-dispersed data, previously checked by dividing the Pearson’s chi-squared statistics with the respective degrees of freedom of the original maximal model with residuals showing a Poisson distribution. The goodness-of-fit of the zero-inflated maximal model was then tested using the Likelihood Ratio Test (LRT [58]) using the lrtest function implemented in the ‘lmtest’ R package [59], which relies on comparing the zero-inflated maximal model with a nested GLM [55].

Model simplification was realized starting from the zero-inflated maximal model and then through removing those covariates that did not produce a significant effect on the response variable. In this case, the removal of non-significant explanatory variables was completed considering the findings of both the count (negative binomial distribution with a “log-link” function) and zero-inflation (negative binomial distribution with a “logit-link” function) model components [56]. Model ranking was performed based on the Akaike’s Information Criterion (AIC [60]) and ΔAIC [61,62]. Those models showing ΔAIC < 2 were considered as competitors of the best model [61,62]. In those cases, the selection of the best model was based on the Akaike’s weight (*ωi* [62]). Before being included in the maximal model, to encourage the maximum likelihood estimates of parameters, covariates were standardized using the mutate_at function implemented in the ‘dplyr’ R package [63]. Multicollinearity among covariates was checked through the Variance Inflation Factor (VIF [64]) implemented in the ‘car’ R package [65]. Although high VIF values do not discount the results of regression analyses [66], we considered a VIF ≥ 10 [67,68,69,70] as a threshold value to define the presence of covariates potentially presenting severe collinearity issues.

## 3. Results

Within the eradication area, ‘Agriculture’ represents 83.4% of the overall surface, followed by ‘Urban areas’ (7.6%) and ‘Wetlands and water bodies’ (3.8%) (Table 1). Conversely, within the non-eradication area, ‘Broad-leaved forests’ covers 45.8% of the overall surface, followed by ‘Coniferous forests’ (15.1%) and ‘Agriculture’ (10.0%) (Table 1).

Within the eradication area, there was a significant difference (χ^2^ = 121.9, *p* < 0.001) in the number of wild boar hunted when comparing 2019 with 2021 and 2020 with 2021 (pnif, *p* < 0.001) (Table 2). There was a significant difference (χ^2^ = 94.6, *p* < 0.001) in wild boar abundance when comparing 2020 with 2021 and 2020 with 2022 (pnif, *p* < 0.001) (Table 2). There was a significant difference (χ^2^ = 9.7, *p* = 0.007) in the number of official claims when comparing 2020 with 2021 (pnif, *p* = 0.001) (Table 2).

Within the non-eradication area, there was a significant difference (χ^2^ = 854.1, *p* < 0.001) in the number of wild boar hunted across years (i.e., 2019 vs. 2020, 2019 vs. 2021, 2020 vs. 2021) (pnif, *p* < 0.001) (Table 2). There was a significant difference (χ^2^ = 263.7, *p* < 0.001) in wild boar abundance across years (i.e., 2019 vs. 2020, 2019 vs. 2021, 2020 vs. 2021) (pnif, *p* < 0.001) (Table 2). There was a significant difference (χ^2^ = 9.1, *p* = 0.01) in the number of official claims when comparing 2020 with 2022 and 2021 with 2022 (pnif, *p* = 0.02) (Table 2).

With regard to the eradication area, from 2020 to 2022, nine crop categories were damaged, i.e., barley *Hordeum* spp., corn *Zea* spp., grassland, lucerne *Medicago* spp., protein pea *Lathyrus* spp., soy *Glycine* spp., sunflower *Helianthus* spp., vineyard *Vitis* spp., wheat *Triticum* spp. There was a significant difference (Fisher’s exact test, *p* < 0.001) in the percentage of damages among affected crops, with ‘corn’ being significantly more damaged (pnif, *p* < 0.001) than other crop categories in all years (Figure 2).

With regard to the non-eradication area, from 2020 to 2022, 18 crop categories were damaged, i.e., apple *Malus* spp., barley, corn, courgette *Cucurbita* spp., grassland, kidney bean *Phaseolus* spp., lavender *Lavandula* spp., leek *Allium* spp., lettuce *Lactuca* spp., lucerne, melon *Cucumis* spp., potato *Solanum* spp., radish *Cichorium* spp., pumpkin *Cucurbita* spp., soy, sunflower, vineyard, wheat. From 2020 to 2022, there was a significant difference (Fisher’s exact test, *p* < 0.001) in the percentage of damages among affected crops, with ‘corn’ and ‘grassland’ being significantly more damaged (pnif, *p* < 0.05) than other crop categories in all years (Figure 2). In 2020, there was a significantly higher (pnif, *p* < 0.05) percentage of damages in ‘potato’ compared to ‘courgette’, ‘kidney bean’, ‘melon’, ‘apple’, ‘pumpkin’ and ‘soy’ (Figure 2). In 2022, there was a significantly higher (pnif, *p* < 0.05) percentage of damages in ‘vineyard’ compared to ‘soy’, ‘sunflower’ and ‘wheat’ (Figure 2).

### 3.1. Spatio-Temporal Distribution of Wild Boar Hunting, Density and Damages and Hunting Effect

#### 3.1.1. Eradication Area

In 2019, single (hot)spots of wild boar hunting density were observed in the south-eastern area of the FVG Region (Figure 3a). In 2020, both a hotspot cluster and a single spot of wild boar density were observed in the south-eastern and southern area, respectively, of the region (Figure 3b). In 2020, a hotspot cluster of official-claims density (Figure 3c) was observed in the south-eastern area of the region.

The first GLM revealed that to one unit increase of wild boar hunted in 2019, a significant increase (β = 0.07, SE = 8 × 10^−3^, *p* < 0.001) in wild boar abundance was observed in 2020. The results obtained from the best ZIMs (a,b in Table 3) revealed that the log odds to observe a significant increasing number of damages (official claims) in 2020 was associated with ‘arboriculture’ (β = 3.5, SE = 0.7, *p* < 0.001) and ‘orchards, vineyards and olive groves’ (β = 0.7, SE = 0.3, *p* = 0.01). Conversely, the log odds to observe a significant decreasing number of damages in 2020 was associated with ‘intensive cultivations’ (β = −2.2, SE = 0.5, *p* < 0.001). Furthermore, the wild boar hunting in 2019 was linked to a significant decrease (β = −5.8, SE = 2.6, *p* = 0.03) in the log odds probability to observe the absence of damages in 2020. Conversely, the wild boar abundance in 2020 was associated with a significant increase (β = 3.1, SE = 1.5, *p* = 0.04) in the log odds probability to observe the absence of damages in 2020.

In 2020, single (hot)spots of wild boar hunting density were observed in the southern and south-eastern area of the region (Figure 4a). In 2021, both hotspot clusters and single (hot)spots of wild boar density were observed in the southern and south-eastern area, respectively, of the region (Figure 4b). Furthermore, a hotspot of official-claims density was observed in the south-eastern area of the region (Figure 4c).

The second GLM revealed that to one unit increase of wild boar hunted in 2020, a significant increase (β = 0.05, SE = 7 × 10^−3^, *p* < 0.001) in wild boar abundance was observed in 2021. The results obtained from the best ZIMs (c,d in Table 3) showed that the log odds to observe a significant increasing number of damages (official claims) in 2021 was associated with ‘arboriculture’ (β = 1.3, SE = 0.7, *p* = 0.04) and ‘orchards, vineyards and olive groves’ (β = 1.3, SE = 0.6, *p* = 0.01). Conversely, the log odds to observe a significant decreasing number of damages in 2021 was associated with ‘intensive cultivations’ (β = −0.9, SE = 0.4, *p* = 0.02). Moreover, the wild boar hunting in 2020 was linked to a significant increase (β = 0.6, SE = 0.3, *p* = 0.01) in the log odds probability to observe an increasing number of damages in 2021. Conversely, the wild boar abundance in 2021 was associated with a significant decrease (β = −0.7, SE = 0.3, *p* = 0.03) in the log odds probability to observe an increasing number of damages in 2021.

In 2021, both hotspot clusters and single (hot)spots of wild boar hunting density were observed in the south-eastern and southern area of the region, respectively (Figure 5a). In 2022, both hotspot clusters and single (hot)spots of wild boar density were observed in the south-eastern and central area, respectively, of the region (Figure 5b). In 2022, a hotspot cluster of official-claims density was observed in the south-eastern area of the region (Figure 5c).

The third GLM revealed that to one unit increase of wild boar hunted in 2021, a significant increase (β = 0.03, SE = 4 × 10^−3^, *p* < 0.001) in wild boar abundance was observed in 2022. The results obtained from the best ZIMs (e,f in Table 3) revealed that the log odds to observe a significant decreasing number of damages (official claims) in 2022 was associated with ‘intensive cultivations’ (β = −1.5, SE = 0.5, *p* = 0.002). In addition, the wild boar abundance in 2022 was associated with a significant decrease (β = −0.9, SE = 0.4, *p* = 0.04) of the log odds probability to observe an increasing number of damages in 2022.

#### 3.1.2. Non-Eradication Area

In 2019, single (hot)spots of wild boar hunting density were observed in the south-eastern area of the region (Figure 6a), and, in 2020, both a hotspot cluster and single (hot)spots of wild boar density were observed in the same area (Figure 6b). In 2020, single (hot)spots of official claims were observed in the eastern and south-eastern area of the region (Figure 6c).

The first GLM revealed that to one unit increase of wild boar hunted in 2019, a significant increase (β = 0.01, SE = 1 × 10^−3^, *p* < 0.001) in wild boar abundance was observed in 2020. The results obtained from the best ZIMs (a,b in Table 4) revealed that the log odds to observe a significant increasing number of damages (official claims) in 2020 were associated with ‘intensive cultivations’ (β = 0.4, SE = 0.09, *p* < 0.001). Furthermore, a non-significant effect of the covariate ‘wild boar hunted in 2019’ was observed for both the count (β = 0.08, SE = 0.13, *p* = 0.53) and zero-inflation (β = −2.4, SE = 1.3, *p* = 0.07) components.

In 2020, both a hotspot cluster and a single (hot)spot of wild boar hunting density were observed in the south-eastern area of the region (Figure 7a), and, in 2021, hotspot clusters of wild boar density were observed in the same area (Figure 7b). In 2021, single (hot)spots of official-claims density were observed in the eastern and south-eastern areas of the region (Figure 7c).

The second GLM revealed that to one unit increase of wild boar hunted in 2020, a significant increase (β = 0.02, SE = 2 × 10^−3^, *p* < 0.001) in wild boar abundance was observed in 2021. The results obtained from the best ZIMs (c,d in Table 4) showed that the log odds to observe a significant absence of damages (official claims) in 2021 were associated with ‘arboriculture’ (β = 22.1, SE = 11.0, *p* = 0.04). Conversely, the log odds to observe a significant presence of damages in 2021 were associated with ‘intensive cultivations’ (β = −49.1, SE = 21.9, *p* = 0.02). Moreover, no significant effects of the covariates ‘wild boar hunted in 2020’ and ‘wild boar abundance in 2021’ were observed for both the count (β_hunted2020_ = −0.1, SE = 0.3, *p* = 0.7; β_abundance2021_ = 0.3, SE = 0.3, *p* = 0.3) and zero-inflation (β_hunted2020_ = −15.8, SE = 9.1, *p* = 0.08; β_abundance2021_ = 1.2, SE = 2.2, *p* = 0.6) components.

In 2021, both a hotspot cluster and a single (hot)spot of wild boar hunting density were observed in the south-eastern area of the region (Figure 8a), and a hotspot cluster of wild boar density was observed in 2022 in the same area (Figure 8b). In 2022, both a hotspot cluster and a single (hot)spot of official claims density were observed in the eastern and northern area of the region, respectively (Figure 8c).

The third GLM revealed that to one unit increase of wild boar hunted in 2021, a significant increase (β = 0.008, SE = 8 × 10^−3^, *p* < 0.001) in wild boar abundance was observed in 2022. The results obtained from the best ZIMs (e,f in Table 4) revealed that the log odds to observe a significant increasing number of damages (official claims) in 2022 were associated with ‘extensive cultivations’ (β = 0.5, SE = 0.1, *p* < 0.001) and ‘intensive cultivations’ (β = 0.4, SE = 0.1, *p* < 0.001). Conversely, the log odds to observe a significant decreasing number of damages in 2021 were associated with ‘grasslands’ (β = −0.6, SE = 0.1, *p* < 0.001). In addition, a non-significant effect of the covariate ‘wild boar hunted in 2021’ was observed for both the count (β = 0.2, SE = 0.2, *p* = 0.3) and zero-inflation (β = −4.0, SE = 3.7, *p* = 0.3) components.

## 4. Discussion

In highly human-altered environments an effective management plan aimed at maintaining the wild boar population up to a sustainable level to decrease the negative interactions with human activities becomes a priority. Although the wild boar is a native species in Europe, where it becomes highly abundant (like in Italy [20,21]) it may exert negative impacts on agricultural activities (e.g., [24,27,29]) and biodiversity in the form of trampling and predation on invertebrates, small vertebrates and eggs of ground-nesting birds (e.g., [71,72,73,74]), especially within protected areas (e.g., Parks or Natura 2000 areas [29]). Furthermore, through rooting behaviour, they can damage plant communities [75,76] and alter soil compositions [77,78].

Within the eradication area, the higher percentage of damages recorded in corn fields is in accordance with the findings reached in other studies (e.g., [24,29]). Corn represents one of the most damaged crops in Italy (e.g., [24,29]) and other European countries (e.g., Spain [23], Luxembourg [26]). Damages to corn are mainly concentrated from June to September, the period that coincides with the sowing and ripening of cobs [29,79], and during which other high-caloric food resources (e.g., acorns) are absent in wooded areas [22]. Within the eradication area, agriculture covers 83.4% of the overall surface, and arboriculture is mostly oriented towards broad-leaved cultivations. The association between agricultural damages and increasing surface of arboriculture may be explained by the use of these habitats by wild boar because of the presence of mast (e.g., [28,80,81,82]) and refuges (e.g., [29,83,84,85]). Therefore, the likelihood of damages increases in those cultivars located in the near proximity of them [29]. As for orchards, vineyards and olive groves, wild boar are known to use these habitats for foraging [27]. Although in the eradication area only few damages were reported towards this crop category, the proximity of certain crops (such as corn) to orchards, vineyards and olive grove fields could affect the occurrence of damages. The likelihood of damages was also negatively associated with the increasing surface of intensive cultivations. Within the eradication area, canopy-covered habitats (i.e., broad-leaved forests, coniferous forests, mixed forests, moors, shrublands and riparian vegetation) constitute 2% of the overall surface, show a patchily distribution and/or are present along the main rivers. Because forest areas provide food (e.g., [28,80,81,82]) and refuges to wild boar (e.g., [29,83,84,85]), we speculate that the probability of damages is higher in those crops located in the near proximity of these habitats, rather than in open fields in which vegetation covers are absent or barely present.

The hotspot of official-claims density observed from 2020 to 2022 within the eradication area, may be explained by the high presence of agricultural fields in the near proximity to rivers and canopy-covered areas. Rivers are used by wild boar as ecological corridors to move across agricultural fields, while forests are used as refuges by the species (e.g., [29,83,84,85]). Therefore, the risk of damages increases in those cultivated fields located in the near proximity of these habitats [29]. A partial spatial overlap among wild boar hunting density, wild boar density and official-claims density was observed only comparing the official claims recorded in 2020 with both wild boar hunting in 2019 and wild boar density in 2020. No clear spatial overlap was observed in the other years. In addition, the results showed that the current level of harvesting per year has led to an increase in terms of wild boar population in the following year. This result may find explanation in the high reproductive potential of the species triggered by a density-dependent compensation effect across time [3,24]. Moreover, especially for the number of official claims reported in 2020 and 2021, the current level of harvesting was associated with an increasing number and/or presence of damages. This finding may be explained by the potential effect of hunting which, in certain contexts, may alter the behaviour of wild boar leading to an increase in their home ranges [86] and higher use of cultivated fields [87,88].

Contrary to our expectations, an inverse relationship was observed between number of damages and wild boar population size. This result contrasts to what is observed in other studies (e.g., [26,89,90]). However, there are other factors apart from wild boar density that are not evaluated in this research, and that determine the likelihood of a particular crop to be damaged or not, e.g., age and social structure of wild boar populations [91], abundance of supplementary feeding points [89], distance of agricultural fields to the nearest forest area (e.g., [29,83,84]) and type of cultivated crop [92]. The latter is of particular importance given that, in FVG, the surface of certain cultivated fields (e.g., corn) may change consistently from one year to another.

Within the non-eradication area, the higher percentage of damages recorded in corn fields and grasslands is in accordance with other research [24,29]. Corn fields are known to be consumed by wild boar (e.g., [23,26]), while, as for grasslands, damages are linked to wild boar trampling and rooting behaviour [77,93] searching for earthworms and invertebrates’ larvae in the upper layers of the soil [94]. Higher number of damages were associated with an increasing surface of arboriculture and both intensive and extensive cultivations, except for 2021 in which damages were negatively associated with the increasing surface of intensive cultivations. Furthermore, in 2022, an inverse relationship was observed between the number of damages and increasing grassland surface. As for arboriculture, the wild boar positively selects these habitats due to the presence of refuges (e.g., [29,83,84,85]) and food (e.g., [28,80,81,82]). Therefore, crop damages are likely to occur when crop fields occur nearby [29]. The same considerations apply to grasslands and both intensive and extensive cultivations, with the probability of damages linked to the near proximity of protective forest areas [29]. Therefore, case scenarios need to be considered.

Within the non-eradication area, from 2020 to 2022, the only hotspot of official claims density was observed in 2022 in the eastern area of the region. In previous years, only single (hot)spots of official claims were observed. Moreover, no clear spatial overlap was observed among hotspot cluster and single (hot)spots of official claims density, and hotspot clusters and single (hot)spots of wild boar hunting and wild boar densities. Within the non-eradication area, agriculture and grasslands constitute 15.9% of the overall surface, while canopy-covered areas cover 66.2% of the overall surface. Because in those municipalities characterized by a higher number of damages agricultural fields and grasslands are fragmented with forest habitats (which constitute the highest percentage of the overall available surface—mean ± SD = 79.4 ± 15.8%), the likelihood of damages increases [29]. The higher wild boar abundance recorded within the non-eradication area compared to the eradication one is most likely attributable to the presence of a more natural and suitable habitat for the species, especially forests (e.g., [28,29,80,81,82,83,84,85,95]), which constitute one of the main factors which favoured the expansion of the wild boar population across Europe (e.g., [2,3]). However, in the eradication area, the wild boar culling system does not strictly depend on local counts. Therefore, the latter may be less accurate compared to those conducted in the non-eradication area. The findings revealed that, also in the non-eradication area, the current level of harvesting per year has led to an increase in terms of wild boar population in the following year, which can be further explained by the reproductive potential of the species triggered by a density-dependent compensation effect over time [3,24]. Furthermore, no significant relationship was found between the number of official claims and both wild boar hunting and abundance. In FVG, the wild boar hunting system prescribes priority hunting of young wild boar (under one year of age) to reduce dispersal and to control (in the case of the non-eradication area) or reduce (in the case of the eradication area) the population size [38]. Although lethal control of wild boar populations is effective to reduce agricultural damages in other countries (e.g., Switzerland [31], Spain [37]), findings presented in this research are consistent with those obtained by Cappa et al. (2021) [24], revealing that the current hunting system carried out in both the eradication and non-eradication area of the FVG region did not significantly reduce the amount of damages. Nevertheless, these results need to be cautiously interpreted considering the small period analysed. Further research involving more data is needed to provide detailed insights.

## 5. Conclusions and Management Implications

To the best of our knowledge, this study represents the first attempt to evaluate the effectiveness of the hunting system in the north-east of Italy in terms of agricultural damages. However, the research presents some limitations imposed by the data set provided by the FVG region: (*i*) the small period analysed allowed to properly explore the effect of hunting only in the short-term; (*ii*) the absence of information regarding the coordinates in which each instance of damage occurred, as well as the surface covered by each crop category in each farm, prevented to explore eventual preferences shown by wild boar towards certain crops in relation to their local availability, as well as the distance of each damage from the nearest forest habitat, thus ascertaining the potential role of woodlands as “refuge areas” for wild boar (especially during the hunting season); (*iii*) because damages towards certain crop categories (e.g., corn) depends on their milky stages, the absence of the date in which damages occurred prevented detailed explanations about the seasonal distribution of damages; (*iv*) the absence of the date in which wild boar were hunted prevented the exploration of the potential inter-annual effect of hunting on the number of agricultural damages. All these limitations suggest that further studies with more accurate data are strongly recommended to cover these aspects in detail.

The findings presented suggested that, within both the eradication and non-eradication area, the current level of harvesting did not lead to a significant reduction of the amount of damages to agricultural fields. Furthermore, harvesting coincided with an increase in wild boar population size across years. The ongoing climate change could optimize the living conditions of wild boar through directly influencing the abundance of natural food resources and agricultural techniques. Therefore, when the species is favoured by these conditions and becomes highly abundant, to effectively reduce crop damages across time, a multifaceted approach involving the numeric control of the species based on accurate counts in parallel carried out with non-lethal prevention measures should be considered. The hunting system in FVG is mainly oriented towards hunting of young individuals. However, because of the high reproductive rate of the species, increasing the mortality rate in mainly (or only) one age class may be ineffective in limiting the population growth. Conversely, a hunting plan focusing on both adults (reproductive females) and juveniles in a similar percentage could represent a valuable solution. However, wild boar numerical control, especially in forests, should be carefully planned to avoid drastic population decline as wild boar, in its native distribution range, exerts important ecological roles that need to be preserved.

## Figures and Tables

**Figure 1 animals-14-00042-f001:**
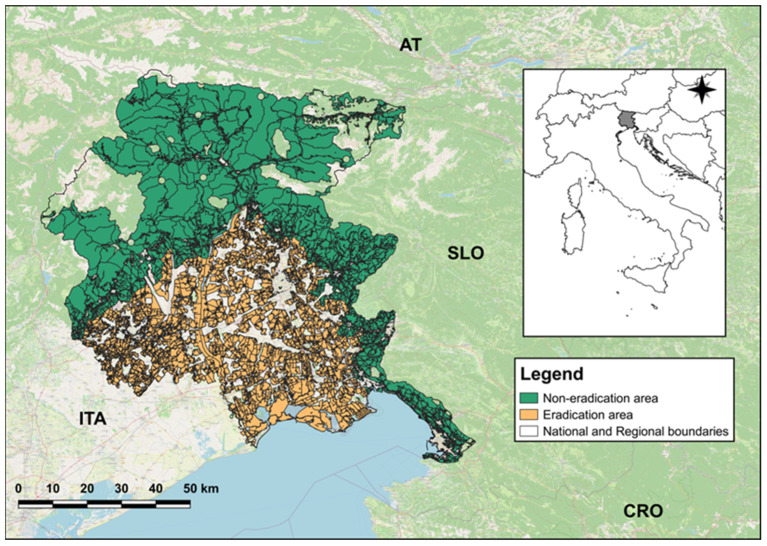
Location of the study area (Friuli Venezia Giulia Region—inset map) divided between eradication and non-eradication areas. Uncoloured polygons identify those territories in which hunting is forbidden. Abbreviations: ITA = Italy; AT = Austria; SLO = Slovenia; CRO = Croatia.

**Figure 2 animals-14-00042-f002:**
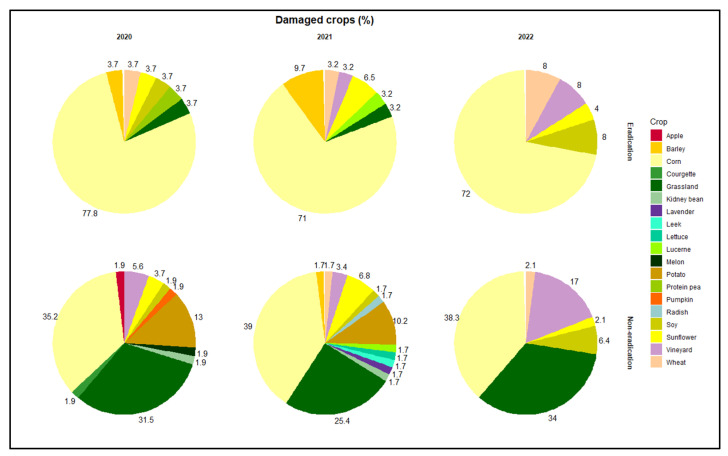
Percentage of crops damaged by wild boar from 2020 to 2022 within the eradication and non-eradication area.

**Figure 3 animals-14-00042-f003:**
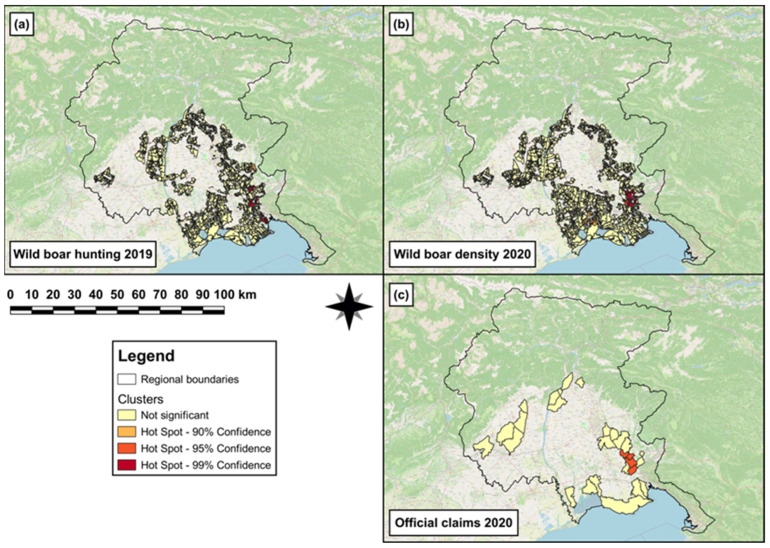
Map showing (**a**) single (hot)spots of wild boar hunting density in 2019 in the hunting areas falling within the eradication area; (**b**) hotspot cluster and single (hot)spot of wild boar density in 2020 in the hunting areas falling within the eradication area; and (**c**) hotspot cluster of official-claims density reported in 2020 in those municipalities falling within the eradication area.

**Figure 4 animals-14-00042-f004:**
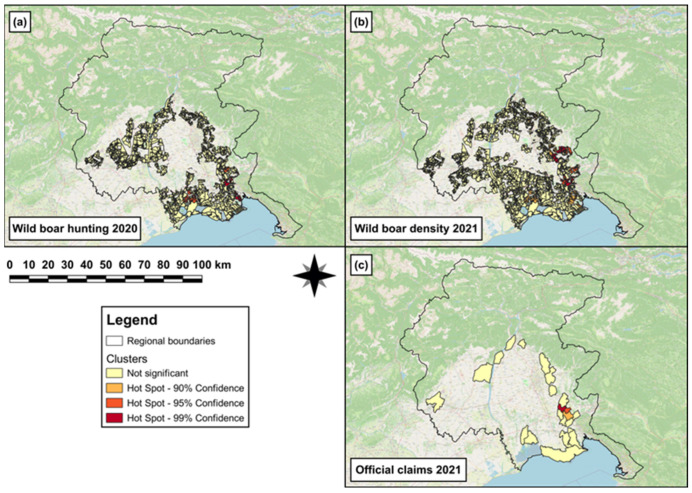
Map showing (**a**) single (hot)spots of wild boar hunting density in 2020 in the hunting areas falling within the eradication area; (**b**) hotspot clusters and single (hot)spots of wild boar density in 2021 in the hunting areas falling within the eradication area; and (**c**) hotspot cluster of official-claims density reported in 2021 in those municipalities falling within the eradication area.

**Figure 5 animals-14-00042-f005:**
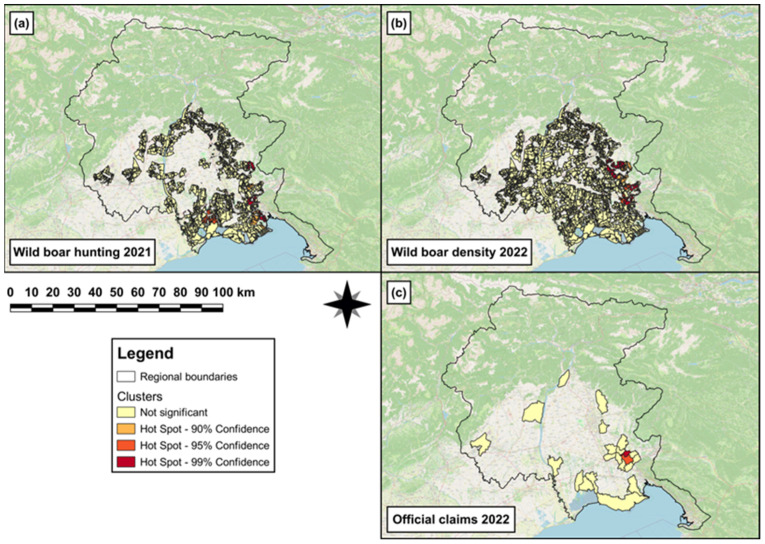
Map showing (**a**) both hotspot clusters and single (hot)spots of wild boar hunting density in 2021 in the hunting areas falling within the eradication area; (**b**) hotspot clusters and single (hot)spots of wild boar density in 2022 in the hunting areas falling within the eradication area; and (**c**) hotspot cluster of official-claims density reported in 2022 in those municipalities falling within the eradication area.

**Figure 6 animals-14-00042-f006:**
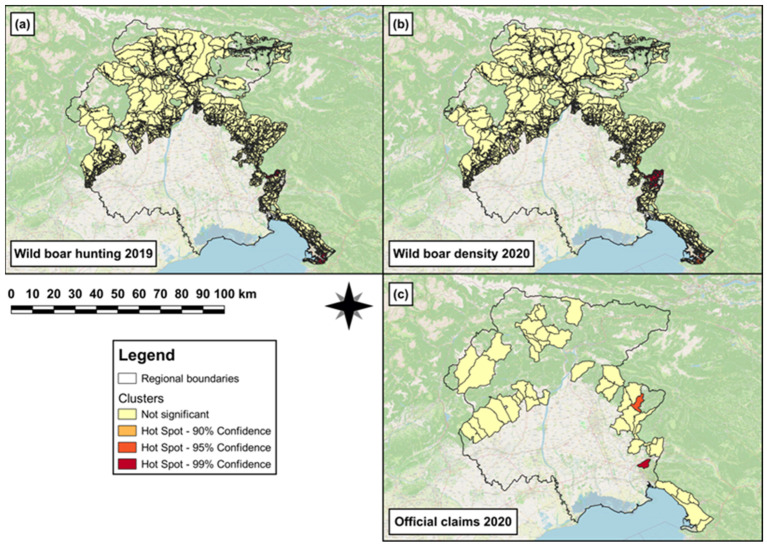
Map showing (**a**) single (hot)spots of wild boar hunting density in 2019 in the hunting areas falling within the non-eradication area; (**b**) hotspot cluster and single (hot)spots of wild boar density in the hunting areas falling within the non-eradication area and (**c**) single (hot)spots of official-claims density reported in 2020 in those municipalities falling within the non-eradication area.

**Figure 7 animals-14-00042-f007:**
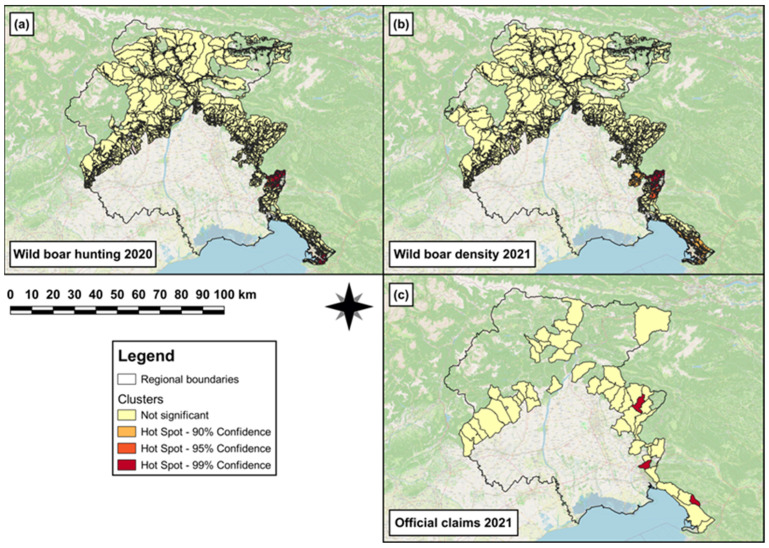
Map showing (**a**) hotspot cluster and single (hot)spot of wild boar hunting density in 2020 in the hunting areas falling within the non-eradication area; (**b**) hotspot clusters of wild boar density in 2021 in the hunting areas falling within the non-eradication area; and (**c**) single (hot)spots of official-claims density reported in 2021 in those municipalities falling within the non-eradication area.

**Figure 8 animals-14-00042-f008:**
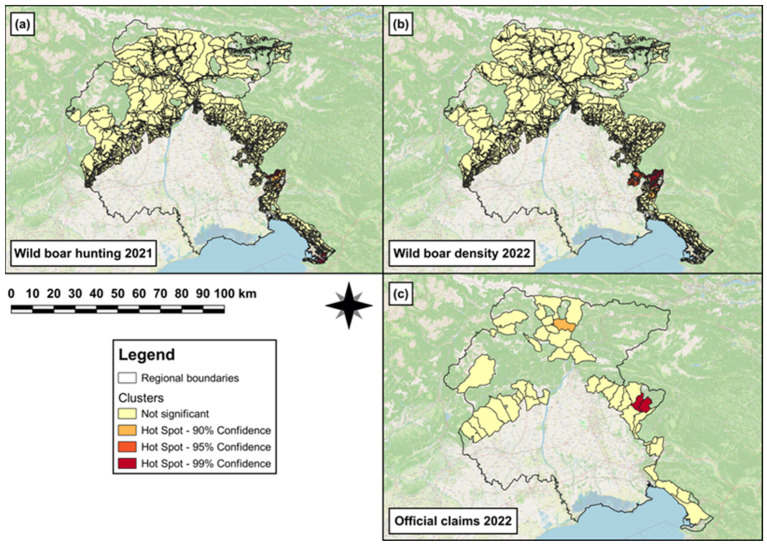
Map showing (**a**) hotspot cluster and single (hot)spot of wild boar hunting density in 2021 in the hunting areas falling within the non-eradication area; (**b**) hotspot cluster of wild boar density in 2022 in the hunting areas falling within the non-eradication area; and (**c**) hotspot cluster and single (hot)spot of official-claims density reported in 2022 in those municipalities falling within the non-eradication area.

**Table 1 animals-14-00042-t001:** Percentage of landscape (PLAND) calculated in each area and for each habitat reclassified, starting from FVG ‘Carta Natura’ 2021 [47].

Area	Habitat	PLAND
Eradication	Agriculture	83.4
	Broad-leaved forests	1.1
	Coniferous forests	0.0
	Dunes, rocky and coastal areas	2.3
	Grasslands	0.9
	Mixed forests	0.1
	Moors, shrublands and riparian vegetation	0.8
	Urban areas	7.6
	Wetlands and water bodies	3.8
Non-eradication	Agriculture	10.0
	Broad-leaved forest	45.8
	Coniferous forests	15.1
	Dunes, rocky/glaciers and coastal areas	6.5
	Grasslands	5.9
	Mixed forests	2.8
	Moors, shrublands and riparian vegetation	2.5
	Urban areas	4.0
	Wetlands and water bodies	7.4

**Table 2 animals-14-00042-t002:** Wild boar hunting (i.e., number of wild boar hunted), abundance (i.e., number of wild boar counted) and damages (i.e., number of official claims for compensations) in each area and across years.

Area	Wild Boar	Year			
		2019	2020	2021	2022
Eradication	Hunting	822	828	1163	944
	Abundance	1492	1138	1517	1478
	Damages	169	90	60	69
Non-eradication	Hunting	3165	2184	4075	3166
	Abundance	3629	3283	4408	4148
	Damages	142	184	180	143

**Table 3 animals-14-00042-t003:** Zero-inflated models ranking with the best model marked in italics. Abbreviations: A = Arboriculture; EC = Extensive cultivations; G = Grasslands; IC = Intensive cultivations; OVOG = Orchards, vineyards and olive groves; Hunting2019/2020/2021 = Wild boar hunted in 2019/2020/2021; Abundance2020/2021/2022 = Wild boar counted in 2020/2021/2022; K = number of coefficients; logLik = log − Likelihood; AIC = Akaike’s information criterion; *ωi* = Akaike’s weight.

Model Section	Model ID	Covariate/s	*K*	*−2logLik*	AIC	ΔAIC	*ωi*
a.	*1*	*A + EC + IC + OVOG*	*10*	*188.1*	*208.1*	*0.0*	*0.7*
	2	A + EC + G + IC + OVOG	12	186.3	210.3	2.2	0.2
	3	A + IC + OVOG	8	202.9	218.9	10.8	0.0
	Null	~1	2	232.9	236.9	28.8	0.0
b.	*1.1*	*Hunting2019 + Abundance2020*	*4*	*198.9*	*206.9*	*0.0*	*1.0*
	Null	~1	2	232.9	236.9	30.0	0.0
c.	*1*	*A + EC + IC + OVOG*	*10*	*164.2*	*184.2*	*0.0*	*0.7*
	2	A + IC + OVOG	8	171.0	187.0	2.8	0.2
	3	A + EC + G + IC + OVOG	12	163.6	187.6	3.4	0.1
	Null	~1	2	191.3	195.3	11.1	0.0
	5	IC	4	188.1	196	11.9	0.0
	6	IC + OVOG	6	187.8	200	15.5	0.0
d.	*1.1*	*Hunting2020 + Abundance2021*	*4*	142.2	*150.2*	*0.0*	*1.0*
	Null	~1	2	191.3	195.3	45.1	0.0
e.	*1*	*A + IC + OVOG*	*8*	*180.1*	*196.1*	*0.0*	*0.3*
	2	A + EC + G + IC + OVOG	12	172.4	196.4	0.4	0.3
	3	IC + OVOG	6	185.8	197.8	1.8	0.1
	4	A + EC + IC + OVOG	10	178.2	198.2	2.2	0.1
	5	IC	4	191.1	199.1	3.0	0.1
	Null	~1	2	196.6	200.6	4.6	0.0
f.	*1.1*	*Hunting2021 + Abundance2022*	*6*	*166.1*	*178.1*	*0.0*	*0.6*
	2.1	Abundance2022	4	170.8	178.8	0.6	0.4
	Null	~1	2	196.6	200.6	22.5	0.0

**Table 4 animals-14-00042-t004:** Zero-inflated models ranking with the best model marked in italics. Abbreviations: A = Arboriculture; EC = Extensive cultivations; G = Grasslands; IC = Intensive cultivations; OVOG = Orchards, vineyards and olive groves; Hunting2019/2020/2021 = Wild boar hunted in 2019/2020/2021; Abundance2020/2021/2022 = Wild boar counted in 2020/2021/2022; K = number of coefficients; logLik = log − Likelihood; AIC = Akaike’s information criterion; *ωi* = Akaike’s weight.

Model Section	Model ID	Covariate/s	*K*	*−2logLik*	AIC	ΔAIC	*ωi*
a.	*1*	*IC + OVOG*	*6*	*301.5*	*313.5*	*0.0*	*0.7*
	2	A + IC + OVOG	8	299.7	315.7	2.2	0.2
	3	IC	4	309.2	317.2	3.7	0.1
	4	A + EC + IC + OVOG	10	301.1	321.1	7.5	0.0
	5	A + EC + G + IC + OVOG	12	300.1	324.1	10.6	0.0
	Null	~1	2	348.6	352.6	39.0	0.0
b.	*1.1*	*Hunting2019*	*4*	*338.3*	*346.3*	*0.0*	*0.8*
	2.1	Hunting2019 + Abundance2020	6	337.8	349.8	3.5	0.1
	Null	~1	2	348.6	352.6	6.3	0.0
c.	*1*	*A + EC + G + IC + OVOG*	*12*	*288.6*	*312.6*	*0.0*	*1.0*
	2	A + EC + G + IC	10	305.7	325.7	13.1	0.0
	3	IC	4	318.2	326.2	13.6	0.0
	4	A + G + IC	8	313.3	329.3	16.7	0.0
	5	A + IC	6	317.6	329.6	17.1	0.0
	Null	~1	2	357.4	361.4	48.8	0.0
d.	*1.1*	*Hunting2020 + Abundance2021*	*6*	*327.6*	*339.6*	*0.0*	*0.9*
	2.1	Abundance2021	4	337.3	345.3	5.6	6 × 10^−2^
	Null	~1	2	357.4	361.4	21.8	2 × 10^−5^
e.	*1*	*EC + G + IC*	*8*	*280.6*	*296.6*	*0.0*	*0.8*
	2	EC + G + IC + OVOG	10	279.9	299.9	3.3	0.2
	3	A + EC + G + IC + OVOG	12	278.3	302.3	5.7	0.0
	Null	~1	2	319.1	323.1	26.6	0.0
f.	*1.1*	*Hunting2021*	*4*	*308.4*	*316.4*	*0.0*	*0.8*
	2.1	Hunting2021 + Abundance2022	6	307.4	319.4	3.0	0.2
	Null	~1	2	319.1	323.1	6.7	0.0

## Data Availability

The data presented in this study were shared by the Friuli Venezia Giulia Region. Data are available on reasonable request from the corresponding author only in the case of previous consent obtained from the Hunting Service and Fishing Resources of the Friuli Venezia Giulia Region. Data are not publicly available due to privacy reasons.

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
