# Peer review of "The Effect of the Wild Boar Hunting System on Agricultural Damages: The North-East of Italy as a Case Scenario"

_animals, 2023, doi:10.3390/ani14010042_

Round 1

Reviewer 1 Report

Comments and Suggestions for Authors

The manuscript provides interesting information on wild boar hunt and its relation with crop damages. The bibliogrraphic research is very good. These are my comments to improve it.

Line 10. Latin name should not have brackets, following the International Code of Zoological Nomenclature. In all the text.

L12. Effect. Maybe results is a better word to express the idea.

L13. When talking about censuses the idea is normally monitoring, estimations, counts, but not strictly censuses.

L13. Hunting bags? Hunting results?

L16. … wild boar hunting, wild boar density and damages to agriculture.

L17. If you say that the hunting system is poorly effective, you mean that there are some positive results, even if modest. Is this what you what you want to say?

L18. More age classes in a similar proportion. Before this statement the way hunting is developed, should be explained. Until now, you talk about hunting and now you say culling. Maybe both concepts should be introduced before as even if they use the same methods, they follow different aims: hunting (sport hunting, maintaining the population) and culling (reducing the population). The idea of “similar proportion” consists in 1:1 sex ratio? Or proportion considering the relative abundance (%) of each age class?

L20. Hunting. See previous comment.

L22. Poorly treated. This is a worldwide problem.

L23. Purpose. There are some words missing before to understand the whole sentence.

L24. Non-eradication I think it’s a better  term.

L25. On the extent. What do you mean?

L26. … “the” zero-inflated models…

L29. Did not significantly reduce. This means that it did reduce damages in a modest way?

L30-32. See L18 comment.

L45. I would add the recovery of forests as one of the main reasons, apart from reforestation and human emigration from rural areas.

L53. Concentrated in forested environments. Where? Today wild boars live in towns, this should be underlined.

L56-57. Crop damages mostly occur in periods in which food resources are scant in natural habitats. Not only in these situations. Food can be abundant in the forest (mast production) and damages occur in field crops (maize).

L59. …where hunting is forbidden. In some cases it has been demonstrated that areas with no hunt have higher abundance of wild boars. I attach an article.

L64. (in decline compared to the wild boar population). In decline, simply. Please add some references, like Massei et al.

L67-68. However, studies carried out in Italy are still limited. Studies are limited everywhere; Italy is not a special case. You refer to Italy several time, but it is not necessary to mention the country most of the times. Animals is an international journal, as well as its readers.

L72-73. Extent of agricultural damages. Extent means surface? Would amount be better?

L77-79. Habitat is mainly composed by forests and woodlands, and lowland areas, mainly covered by agriculture. Wild boars also live above the timberline in the Alps, urban environments, marshes and coastal ecosystems. What’s the difference between forests and woodlands?

L80-81. The overall wild boar density across 80 the Region is of about 1.7 ind./100 ha. Does the density refer to forest surface or the surface of hunting grounds? Using official records is OK if data is reliable. An explanation of the methods used to calculate the density is needed, this is very important. In detail.

L90. Is a hunting reserve an area where there is no hunt?

L103. There is no information on the main forest species, which is crucial for the carrying capacity of these habitats for wild boar.

L107. What do you mean for collective hunt? Batues or drives? Explain how it is performed.

L111. (Figure 1 – 38). What is 38? Is it [38]?

L111-113. To the eradication area are associated those municipalities in which cultivated fields are 112 predominant in lowlands and hilly areas, and where wild boar can cause damages. Rephrase.

L120. What is a natural feeding points? Is there artificial feeding?

L127. The way numbers and density of wild boars is calculated must be clarified in detail to make all the rest of results comprehensible.

L135. Protected oasis? Please explain. Protected  Areas should also be mentioned, particularly if hunting is allowed or not inside.

L257. Claims. Explain more in detail what is a claim. The number seems very low for the total study area.

L303. The results of year 2019 is missing in Figure 2. All the Latin names of crops whould appear in the body text.

L555. Wild boar is endemic of the area, not an introduced species. Wild boar is not a problem for biodiversity conservation in general terms. Of course rooting and other feeding activities have an impact in the environment, as any other species impacts the ecosystem where it lives. I would not be so categorical with the impact of wild boars in their natural enviroments.

L562. Acorns. Information on the forest species producing hard mast (beech, oaks) is needed in the study area section.

L630. Avoid giving results in the discussion section.

L656. Put quotations inside […].

L660. Small temporal trend. Three years is not enough to talk about trend.

Comments on the Quality of English Language

English is fine but a review by a native speaker could improve it. I found only some small mistakes along the text.

Author Response

Cover Letter for Scientific Manuscript: R1 (Manuscript ID: animals-2695155)

Dear Reviewer,

We would like to thank you for the time you dedicated to revise our manuscript and for the important suggestions which considerably increased the quality of our work. The responses to your comments were provided in bold down below. Moreover, the changes through the main body text were highlighted in YELLOW to improve clarity.

Thank you. We appreciated your time and we hope that, after the applied revisions, you will find our manuscript of interest and suitable for publications in Animals.

Kind regards,

Marcello Franchini, Ph.D.

Post–Doc Researcher – Department of Agricultural, Food, Environmental and Animal Sciences, University of Udine, Via Delle Scienze 206, Udine, 33100, Italy.

Telephone (mobile): +39 3382117729

E–mail: marcello.franchini@uniud.it

The manuscript provides interesting information on wild boar hunt and its relation with crop damages. The bibliographic research is very good.

Many thanks for this appreciation.

These are my comments to improve it:

Line 10. Latin name should not have brackets, following the International Code of Zoological Nomenclature. In all the text.

Suggestion accepted and implemented within the text.

L12. Effect. Maybe results is a better word to express the idea.

Suggestion accepted and implemented within the text.

L13. When talking about censuses the idea is normally monitoring, estimations, counts, but not strictly censuses.

Suggestion accepted. We used the word “count” to improve clarity.

L13. Hunting bags? Hunting results?

Suggestion accepted. We used the word “hunting bags” to improve clarity.

L16. … wild boar hunting, wild boar density and damages to agriculture.

Suggestion accepted and implemented within the text.

L17. If you say that the hunting system is poorly effective, you mean that there are some positive results, even if modest. Is this what you what you want to say?

What we meant was that the current level of hunting (both in the eradication and non–eradication area) do not lead to a significant increasing or decreasing numbers of wild boar damages to agriculture. We reframed the sentence.

L18. More age classes in a similar proportion. Before this statement the way hunting is developed, should be explained. Until now, you talk about hunting and now you say culling. Maybe both concepts should be introduced before as even if they use the same methods, they follow different aims: hunting (sport hunting, maintaining the population) and culling (reducing the population). The idea of “similar proportion” consists in 1:1 sex ratio? Or proportion considering the relative abundance (%) of each age class?

We thank the referee for this important suggestion. Hunting is the term we used in the abstract to stay within the 200 required words. However, we introduced the difference between hunting (realized within the non-eradication area) and culling (realized within the eradication area) in the paragraph “2.2. Wild boar hunting system in FVG”.

We changed the word “proportion” with “percentage” to improve clarity.

L20. Hunting. See previous comment.

Suggestion accepted and implemented within the text.

L22. Poorly treated. This is a worldwide problem.

Suggestion accepted and implemented within the text.

L23. Purpose. There are some words missing before to understand the whole sentence.

Suggestion accepted. The sentence was reframed to improve clarity.

L24. Non-eradication I think it’s a better term.

Suggestion accepted and implemented within the text (as well as in Figure 1).

L25. On the extent. What do you mean?

As suggested by the referee, we changed the word with “number”.

L26. … “the” zero-inflated models…

Suggestion accepted and implemented within the text.

L29. Did not significantly reduce. This means that it did reduce damages in a modest way?

We changed the word with “affect” to improve clarity.

L30-32. See L18 comment.

Suggestion accepted and implemented within the text. We changed the word “proportion” with “percentage” to improve clarity.

L45. I would add the recovery of forests as one of the main reasons, apart from reforestation and human emigration from rural areas.

Suggestion accepted and implemented within the text.

L53. Concentrated in forested environments. Where? Today wild boars live in towns, this should be underlined.

Suggestion accepted. The sentence was reframed.

L56-57. Crop damages mostly occur in periods in which food resources are scant in natural habitats. Not only in these situations. Food can be abundant in the forest (mast production) and damages occur in field crops (maize).

Suggestion accepted. We reframed the sentence.

L59. …where hunting is forbidden. In some cases it has been demonstrated that areas with no hunt have higher abundance of wild boars. I attach an article.

The meaning of the sentence was to explain that the likelihood of agricultural damages could mostly occur in areas in which hunting is forbidden (like in protected areas). The article you attached reports that non–hunting areas constitute refuges for wild boar, and the species tends to avoid agricultural fields in all seasons (apparently because of seasonal limitations of food availability) instead selecting wooded areas with hard mast production. Therefore, it provides different information compared to what we wanted to highlight. However, because the work provides useful information about the role of forests as refuge areas and sources of food for the wild boar, it was included within the list of references.

L64. (in decline compared to the wild boar population). In decline, simply. Please add some references, like Massei et al.

Suggestion accepted and implemented within the text.

L67-68. However, studies carried out in Italy are still limited. Studies are limited everywhere; Italy is not a special case. You refer to Italy several time, but it is not necessary to mention the country most of the times. Animals is an international journal, as well as its readers.

Suggestion accepted. We reframed the sentence.

L72-73. Extent of agricultural damages. Extent means surface? Would amount be better?

Suggestion accepted and implemented within the text. We used the word “number” to improve clarity.

L77-79. Habitat is mainly composed by forests and woodlands, and lowland areas, mainly covered by agriculture. Wild boars also live above the timberline in the Alps, urban environments, marshes and coastal ecosystems. What’s the difference between forests and woodlands?

We agree with the suggestion. We used only the word “forest”.

L80-81. The overall wild boar density across the Region is of about 1.7 ind./100 ha. Does the density refer to forest surface or the surface of hunting grounds? Using official records is OK if data is reliable. An explanation of the methods used to calculate the density is needed, this is very important. In detail.

The density refers to local counts conducted on the agro–sylvo–pastoral surface of each hunting reserve falling within each municipality. The density value was obtained by dividing the overall number of wild boar counted and the agro–sylvo–pastoral surface. Detailed explanations were provided in lines 113–115, in the paragraph “2.2. Wild boar hunting system in FVG” as well as in lines 169–171.

L90. Is a hunting reserve an area where there is no hunt?

No. It’s an area in which hunting is realized. We added a sentence to improve clarity.

L103. There is no information on the main forest species, which is crucial for the carrying capacity of these habitats for wild boar.

Suggestion accepted. A description of the main vegetal species was provided in the study area section.

L107. What do you mean for collective hunt? Battues or drives? Explain how it is performed.

The hunting system is carried out mainly through driven hunting using hunting hounds. A sentence was introduced to improve clarity.

L111. (Figure 1 – 38). What is 38? Is it [38]?

38 is the citation referring to the Wildlife Regional Hunting Plan (2015). The sentence was rephrased to improve clarity.

L111-113. To the eradication area are associated those municipalities in which cultivated fields are predominant in lowlands and hilly areas, and where wild boar can cause damages. Rephrase.

Suggestion accepted and implemented within the text.

L120. What is a natural feeding points? Is there artificial feeding?

No. The sentence is correct. As reported in the Wildlife Regional Hunting Plan (2015), a natural feeding point is an area (open or in the near proximity of forests) in which hunters have commonly seen wild boar feeding during local counts conducted in each hunting reserve.

L127. The way numbers and density of wild boars is calculated must be clarified in detail to make all the rest of results comprehensible.

As for the wild boar number, a sentence was reported in lines 113–115 in which the relationship between local counts and agro–sylvo­–pastoral surface is explained. Conversely, as for the wild boar density calculation a sentence was reported in lines 169–171.

L135. Protected oasis? Please explain. Protected Areas should also be mentioned, particularly if hunting is allowed or not inside.

Protected oases are embedded within protected areas in which hunting is forbidden. We added a sentence in line 157 to improve clarity.

L257. Claims. Explain more in detail what is a claim. The number seems very low for the total study area.

Suggestion accepted. A sentence was introduced to improve clarity.

L303. The results of year 2019 is missing in Figure 2. All the Latin names of crops would appear in the body text.

As for 2019, we excluded the damaged crop categories to be in line with the results presented in the models (i.e., official claims due to crop damages in 2020, 2021, and 2022). As for the Latin names of crops, we accepted the suggestions and added information in lines 305–307 and 312–315.

L555. Wild boar is endemic of the area, not an introduced species. Wild boar is not a problem for biodiversity conservation in general terms. Of course, rooting and other feeding activities have an impact in the environment, as any other species impacts the ecosystem where it lives. I would not be so categorical with the impact of wild boars in their natural environments.

We agree with the suggestion. We introduced a sentence specifying that the wild boar is endemic in Europe and that represents a problem when it became overabundant. This is specifically the case of Italy (we reported some supporting literature). Here, the population is composed also by individuals released (in the past) from a Central European population for hunting purposes (e.g., Mazzoni della Stella et al. 1995). Among others, factors including climatic changes, releases for hunting purposes (as stated above) and absence (or sporadic presence) of natural predators (e.g., the wolf was on the brink of extinction in Italy up to 1971 when it was included among the protected species) have contributed to a notable increase in terms of wild boar population, with consequent negative effects on agriculture and ecosystems.

Mazzoni della Stella R. et al. (1995) The wild boar management in a province of the central Italy. IBEX J.M.E. 3:213–216.

L562. Acorns. Information on the forest species producing hard mast (beech, oaks) is needed in the study area section.

Suggestion accepted. A description of the main vegetal species was provided in the study area section.

L630. Avoid giving results in the discussion section.

Thank you for this suggestion. In this case, as reported in other literatures, we believe that reporting the percentage values could help the reader to better understand the meaning and have a clearer idea about the percentage of each habitat in the study area.

L656. Put quotations inside […].

Suggestion accepted and implemented within the text.

L660. Small temporal trend. Three years is not enough to talk about trend.

Suggestion accepted. We changed the word with “period”.

English is fine but a review by a native speaker could improve it. I found only some small mistakes along the text.

Suggestion accepted. The text was revised by a native English speaker.

Reviewer 2 Report

Comments and Suggestions for Authors

Dear Authors,

all in all your MS is interesting and timely. It is mainly well written and of good quality,

However, I have some concerns about its publication at it recent form.

The Introdcution is quite short. I recommend you give more information on the case scenario and construct an introduction leading to tru hypotheses.

The results section is very long and boring (not hte results themselves, but their description). The text needs to be condensed. Please do not repeat phrases and don´t produce reduntant passages. The results section is written reconditely. Please condense and combine the multiple figures and tables. There is no need to show all single results in detail, better show the main outcomes clearly to be understandable even for scientists out of our small bubble! You may show the single results in the supplement. Usually the results text is not longer than one or two text pages!

The "conclusion" section is quite long for a conclusion.

Fig. 1: I would not name Austria like usually Asutralia is abbreviated (AUS), better just write AT

L87ff: As also some protected areas are included this effect should also be investigated. One variable could be: distance to protected areas (as well as to no erdication)

L108: I would call the "hunting towers" raised hides, which is more usual!

L133: you should name the software here already, not at the end!

L181, L187, L250: all three softwares (languages) need to be cited (there are citation spefications from the software publishers)

L549: don´t use overabundant, here highly abundant would fit better.

L550: delite text in brackets, this is not true! The main reason is increased nutrition due to climate change and agricultural techniques. Predators are not able to stop wb population increase just to slow down when food is highly abundant. => thus, just delete the subset.

L560: No, wild boar use maize in Autumn or summer, but prefer other seeds (e.g. rapeseed). There is also some Literature existing on that, but 'in generell preferring maize' is the wrong wording!

L570: please name the author(s) for [27] (always when directly citing)

652: What is about the time delay?

L680: "all these limitations" => This phrase leads me to one comment: WHen you are not sure about your data and see a lot of limitations => gather better data or better analyses, reanalyse and rewrite the MS!

L705: fertility reduction will not have a proper effect (and do not work up to now), following: acoustic and olfactory deterrents have only seldom an effect without simultaneous hunting, dissuasive feeding does not help and may be counteraproductive. There is a lot of literature showing exactly the oposite to the few you cited!

L695: I recommend to read GETHÖFFER F, KEULING O, MAISTRELLI C, LUDWIG T, SIEBERT U (2023): Heavy Youngsters - Habitat and Climate Factors Lead to a Significant Increase in Body Weight of Wild Boar Females. Animals 13 (5), 898. and think about your conclusions.

Supplement: The information does not deliver any helpful extra information: a waste of storage!

Author Response

Cover Letter for Scientific Manuscript: R1 (Manuscript ID: animals-2695155)

Dear Reviewer,

We would like to thank you for the time you dedicated to revise our manuscript and for the important suggestions which considerably increased the quality of our work. The responses to your comments were provided in bold down below. Moreover, the changes through the main body text were highlighted in YELLOW to improve clarity.

Thank you. We appreciated your time and we hope that, after the applied revisions, you will find our manuscript of interest and suitable for publications in Animals.

Kind regards,

Marcello Franchini, Ph.D.

Post–Doc Researcher – Department of Agricultural, Food, Environmental and Animal Sciences, University of Udine, Via Delle Scienze 206, Udine, 33100, Italy.

Telephone (mobile): +39 3382117729

E–mail: marcello.franchini@uniud.it

Dear Authors,

all in all your MS is interesting and timely. It is mainly well written and of good quality,

Thank you for this appreciation.

However, I have some concerns about its publication at it recent form.

The Introduction is quite short. I recommend you give more information on the case scenario and construct an introduction leading to true hypotheses.

Suggestion accepted. We added more information and both research questions and hypothesis to better explain how to achieve our goal.

The results section is very long and boring (not the results themselves, but their description). The text needs to be condensed. Please do not repeat phrases and don´t produce redundant passages. The results section is written reconditely. Please condense and combine the multiple figures and tables. There is no need to show all single results in detail, better show the main outcomes clearly to be understandable even for scientists out of our small bubble! You may show the single results in the supplement. Usually, the results text is no longer than one or two text pages!

Suggestion accepted. Results were condensed to show the main outcomes and all the redundances were removed. Furthermore, we merged the multiple tables. As for figures, we merged figure 1 and 2 (those referring to crop damages). However, regarding the others, we tried to merge them but the quality was considerably reduced. Therefore, we would prefer to maintain the current format to not compromise the clarity.

The "conclusion" section is quite long for a conclusion.

Suggestion accepted. The conclusions were shortened.

Fig. 1: I would not name Austria like usually Australia is abbreviated (AUS), better just write AT

Suggestion accepted. The Figure 1 was changed accordingly.

L87ff: As also some protected areas are included this effect should also be investigated. One variable could be: distance to protected areas (as well as to no eradication)

We agreed with the referee suggestion. At the beginning we thought to include also the distance of each municipality to the nearest protected area (herein, PAs) in the model, by applying a centroid to each municipality. However, after careful considerations and analyses (both statistical and explorative), we decided to not include it mainly because of three reasons:

  1. The vast majority of PAs are located within the non–eradication area. This led to a disproportion in terms of distances from municipalities located in both the eradication and non–eradication area to PAs, consequently leading to unreliable and hardly comparable results between areas.
  2. As reported in some literatures cited across the main body text (e.g., Welander 2000; Wilson 2004; Linkie et al. 2007; Fonseca 2008; Thurfjell et al. 2009; Amici et al. 2012; Ferens et al. 2023), wild boar is known to use forest habitats because of the presence of food and refuges. Given these considerations, it would have been mostly useful to calculate the distance from the nearest forest cover (abundant in the non–eradication area, while mainly concentrated along the main riparian areas in the eradication area). However, to do so, the coordinates of damages would be needed (all these limits are reported in the “Conclusions”). In the same way, applying a centroid to each municipality and calculating the distance from the nearest canopy–covered areas does not give useful information. In fact, the centroids may fall far from the nearest forest area but damages may have occurred in the near proximity of it.
  3. Zero–inflated models are useful when the response variable presents a high number of zeros. However, they are very sensitive to overfitting depending on the complexity of the model and the size of the dataset. Because we had to deal with a relatively small data set per year, by including several variables, the parameters’ estimation was difficult and a lot of NAs were produced. Therefore, we decided to include only the main habitat variables which, based on the current literature, may significantly influence wild boar damages to cultivars.

L108: I would call the "hunting towers" raised hides, which is more usual!

Suggestion accepted and implemented within the main body text.

L133: you should name the software here already, not at the end!

Suggestion accepted and implemented within the main body text.

L181, L187, L250: all three softwares (languages) need to be cited (there are citation specifications from the software publishers)

Suggestion accepted and implemented within the main body text.

L549: don´t use overabundant, here highly abundant would fit better.

Suggestion accepted and implemented within the main body text.

L550: delete text in brackets, this is not true! The main reason is increased nutrition due to climate change and agricultural techniques. Predators are not able to stop wb population increase just to slow down when food is highly abundant. => thus, just delete the subset.

Suggestion accepted and implemented within the main body text.

L560: No, wild boar use maize in Autumn or summer, but prefer other seeds (e.g. rapeseed). There is also some Literature existing on that, but 'in general preferring maize' is the wrong wording!

Suggestion accepted. We changed with the word “consumed”.

L570: please name the author(s) for [27] (always when directly citing)

Suggestion accepted and implemented within the main body text.

652: What is about the time delay?

According to other suggestions, the sentence was reframed.

L680: "all these limitations" => This phrase leads me to one comment: When you are not sure about your data and see a lot of limitations => gather better data or better analyses, reanalyse and rewrite the MS!

We agree with the referee. As reported in the main body text, the limits are linked to the nature of the dataset we received by the local administrations. Despite these limitations, we considered as important to present these preliminary results not only to add a piece of knowledge about the efficacy of hunting on wild boar agricultural damages (a still poorly treated topic), but also to “send a message” to the local administration about the way in which, from our point of view, data should be collected and shared (please, see the “Conclusions”). A synergistic participation between universities and local administrations is of special importance to collected useful and detailed information about human–wildlife conflicts, in turn allowing to elaborate the most effective management and conservation strategies.

L705: fertility reduction will not have a proper effect (and do not work up to now), following: acoustic and olfactory deterrents have only seldom an effect without simultaneous hunting, dissuasive feeding does not help and may be counterproductive. There is a lot of literature showing exactly the opposite to the few you cited!

Suggestion accepted. The paragraph was reframed.

L695: I recommend to read GETHÖFFER F, KEULING O, MAISTRELLI C, LUDWIG T, SIEBERT U (2023): Heavy Youngsters - Habitat and Climate Factors Lead to a Significant Increase in Body Weight of Wild Boar Females. Animals 13 (5), 898. and think about your conclusions.

Suggestion accepted. The article was read (and cited) and the conclusions were reframed accordingly.

Supplement: The information does not deliver any helpful extra information: a waste of storage!

Suggestion accepted. The old supplementary material was removed.

Reviewer 3 Report

Comments and Suggestions for Authors

Thank you for the article you sent.

In the title of the article it is worth adding "the effect"

There is repeated information in lines 77 and 79/80.

There are only 9 Figures in the publication, why the range of 1-38 in line 111?

What specifically in the results do the links in 155 and 158 lines lead to?

Table 2. in my opinion is missing statistical results.

The results described in section 3.1 are difficult to understand, perhaps there is an opportunity to describe them better?

Literature items 12 and 22 need to be unified journal names.

Author Response

Cover Letter for Scientific Manuscript: R1 (Manuscript ID: animals-2695155)

Dear Reviewer,

We would like to thank you for the time you dedicated to revise our manuscript and for the suggestions which increased the quality of our work. The responses to your comments were provided in bold down below. Moreover, the changes through the main body text were highlighted in YELLOW to improve clarity.

Thank you. We appreciated your time and we hope that, after the applied revisions, you will find our manuscript of interest and suitable for publications in Animals.

Kind regards,

Marcello Franchini, Ph.D.

Post–Doc Researcher – Department of Agricultural, Food, Environmental and Animal Sciences, University of Udine, Via Delle Scienze 206, Udine, 33100, Italy.

Telephone (mobile): +39 3382117729

E–mail: marcello.franchini@uniud.it

Thank you for the article you sent.

In the title of the article, it is worth adding "the effect".

Suggestion accepted and implemented within the text.

There is repeated information in lines 77 and 79/80.

Suggestion accepted. The sentence was reframed.

There are only 9 Figures in the publication, why the range of 1-38 in line 111?

We apology for this misunderstanding. The purpose was to include the citation in quotation brackets. However, because of the confusion, we reframed the sentence.

What specifically in the results do the links in 155 and 158 lines lead to?

At the beginning, the aim was to realize the hotspot maps only including the agricultural surfaces (including grasslands) damaged by wild boar. However, because these surfaces are widely more abundant in the eradication area compared to the non–eradication one, in the latter, the output obtained was hardly visible. Therefore, we calculated the values of official claim density [i.e., overall number of official claims per municipality divided by the surface covered by each habitat potentially damaged by wild boar, i.e., arboriculture (broad–leaved and coniferous plantations), intensive and extensive cultivations, orchards, vineyards and olive groves, grasslands] but, to improve clarity, results were represented using the surfaces covered by the whole municipalities. We added a further sentence.

Table 2. in my opinion is missing statistical results.

I’m afraid we failed to understand the comment. Table 2 reports the overall numbers of wild boar hunted, counted and damages to agriculture. Statistical results were reported in the main body text.

The results described in section 3.1 are difficult to understand, perhaps there is an opportunity to describe them better?

Suggestion accepted. Results were re–written to improve clarity.

Literature items 12 and 22 need to be unified journal names.

Suggestion accepted and implemented within the text.

Reviewer 4 Report

Comments and Suggestions for Authors

Sterilization is not a prevention system applied in the field, but only experimental

as recognized by the authors the sample is too small to draw conclusions, however it remains a correct approach to implement with new data over time. We do not agree with the harm reduction approach (fences, reduction of fertility, etc.) as they either simply move the problem or do not yet have proven scientific effectiveness.

The same applies to the consequences on other species such as deer and roe deer. The thesis is not supported by data

Author Response

Dear reviewer,

We are very grateful for the time you dedicated to revise our manuscript.

Cover Letter for Scientific Manuscript: R1 (Manuscript ID: animals-2695155)

Dear Reviewer,

We would like to thank you for the time you dedicated to revise our manuscript and for the suggestions which increased the quality of our work. The responses to your comments were provided in bold down below. Moreover, the changes through the main body text were highlighted in YELLOW to improve clarity.

Thank you. We appreciated your time and we hope that, after the applied revisions, you will find our manuscript of interest and suitable for publications in Animals.

Kind regards,

Marcello Franchini, Ph.D.

Post–Doc Researcher – Department of Agricultural, Food, Environmental and Animal Sciences, University of Udine, Via Delle Scienze 206, Udine, 33100, Italy.

Telephone (mobile): +39 3382117729

E–mail: marcello.franchini@uniud.it

Sterilization is not a prevention system applied in the field, but only experimental

Suggestion accepted. The paragraph was reframed.

as recognized by the authors the sample is too small to draw conclusions, however it remains a correct approach to implement with new data over time. We do not agree with the harm reduction approach (fences, reduction of fertility, etc.) as they either simply move the problem or do not yet have proven scientific effectiveness.

Suggestion accepted. The paragraph was reframed.

The same applies to the consequences on other species such as deer and roe deer. The thesis is not supported by data.

Suggestion accepted. The sentence was reframed.

Round 2

Reviewer 2 Report

Comments and Suggestions for Authors

I am still not convinced.

The Introduction has improved a bit. The results are still much too long, the conclusions are still an own discussion, not a short conclusion, as it should be.

For convincing the regional authorities, publishing a "report" instead of a scientific sound manuscript in an international journal is not the best way!

The lenght of the publication does not fit it´s scientific content. You just shortened by 1 page! Are you kidding? The length even does not fit your response letter!=> The MS must be reduced by at least 40% (better more than 50%)

Sorry, I like your idea quite a lot, but in my opinion this MS is still far from publication!

Author Response

Dear reviewer,

Thank you again for the time you dedicated to revise our work. Please, see the attached document.

Kind regards,

Marcello Franchini
